# Cooperative enzymatic control of N-acyl amino acids by PM20D1 and FAAH

Joon T Kim[1,2], Stephanie M Terrell[1,2], Veronica L Li[1,2], Wei Wei[1,2,3], Curt R Fischer[2], Jonathan Z Long[1,2]*

[1]Department of Pathology, Stanford University School of Medicine, Stanford, United States; [2]Stanford ChEM-H, Stanford University, Stanford, United States; [3]Department of Biology, Stanford University, Stanford, United States

**Abstract** The N-acyl amino acids are a family of bioactive lipids with pleiotropic physiologic functions, including in energy homeostasis. Their endogenous levels are regulated by an extracellular mammalian N-acyl amino acid synthase/hydrolase called PM20D1 (peptidase M20 domain containing 1). Using an activity-guided biochemical approach, we report the molecular identification of fatty acid amide hydrolase (FAAH) as a second intracellular N-acyl amino acid synthase/hydrolase. In vitro, FAAH exhibits a more restricted substrate scope compared to PM20D1. In mice, genetic ablation or selective pharmacological inhibition of FAAH bidirectionally dysregulates intracellular, but not circulating, N-acyl amino acids. Dual blockade of both PM20D1 and FAAH reveals a dramatic and non-additive biochemical engagement of these two enzymatic pathways. These data establish FAAH as a second intracellular pathway for N-acyl amino acid metabolism and underscore enzymatic division of labor as an enabling strategy for the regulation of a structurally diverse bioactive lipid family.

## Introduction

The N-acyl amino acids are a large family of bioactive lipids composed of a fatty-acid tail conjugated to an amino acid head group. Structurally, N-acyl amino acids are closely related to the other bioactive fatty acid amides including the endogenous cannabinoid receptor agonist anandamide (N-arachidonoylethanolamine) and the N-acyl-homoserine lactone family of bacterial quorum-sensing molecules (*Devane et al., 1992*; *Parsek et al., 1999*). Individual members of the mammalian N-acyl amino acids have been previously implicated in appetite, nociception, vasoregulation, and bone health (*Milman et al., 2006*; *Mostyn et al., 2019*; *Parmar and Ho, 2010*; *Smoum et al., 2010*; *Wu et al., 2017*). We have recently identified a new role for certain N-acyl amino acids in stimulating oxidative metabolism via mitochondrial uncoupling (*Long et al., 2016*). These thermogenic N-acyl amino acids are characterized by medium-chain fatty acyl chains and neutral amino acid head groups, chemical features that are present in a subset of family members including N-acyl phenylalanines, N-acyl leucines, N-acyl glycines, and N-acyl serines (*Keipert et al., 2017*; *Lin et al., 2018*). Consequently, administration of these N-acyl amino acids to mice rendered obese by feeding a high-fat diet increases energy expenditure, reduces adiposity, and improves glucose homeostasis.

Because of the potent effects of N-acyl amino acids on mitochondrial respiration and energy expenditure, the enzymes of N-acyl amino acid biosynthesis and degradation represent candidate pathways for regulating organismal energy homeostasis. To date, only a single mammalian enzyme called PM20D1 (peptidase M20 domain containing 1) has been identified as a physiologic regulator of endogenous N-acyl amino acid levels (*Long et al., 2016*; *Long et al., 2018*). PM20D1 is a classically secreted enzyme present in the circulation in both mice and humans (*Schwenk et al., 2017*; *Uhlén et al., 2015*). In vitro, recombinant PM20D1 functions as a bidirectional N-acyl amino acid synthase/hydrolase, catalyzing the biosynthesis of N-acyl amino acids from free fatty acids and free

*For correspondence:
jzlong@stanford.edu

Competing interests: The authors declare that no competing interests exist.

amino acids, as well as the reverse hydrolysis reaction. Overexpression of PM20D1 in mice, achieved by adeno-associated viral vectors, drives the biosynthesis of circulating N-acyl amino acids in vivo. These mice consequently have increased energy expenditure and reduced adiposity on high-fat diet. Conversely, global ablation of PM20D1 leads to a complex, bidirectional dysregulation of N-acyl amino acid levels and metabolic dysfunction characterized by glucose intolerance and decreased insulin sensitivity. Polymorphisms within and near the human *PM20D1* gene are linked to body mass index (*Benson et al., 2019*; *Bycroft et al., 2018*), providing powerful genetic evidence that PM20D1 may also regulate human obesity and metabolic disorders.

Beyond PM20D1, other mammalian enzymes are also likely to contribute to N-acyl amino acid metabolism, especially considering the large and structurally diverse nature of this lipid family (*Aneetha et al., 2009*; *Bradshaw et al., 2009*; *Cohen et al., 2017*; *Waluk et al., 2010*). To date, the identity of these additional enzymes has remained unknown. Here we use PM20D1-KO tissues to molecularly characterize a second, PM20D1-independent N-acyl amino acid hydrolysis activity. We identify the responsible enzyme as fatty acid amide hydrolase (FAAH) and establish how PM20D1 and FAAH engage in extensive non-additive interactions in vivo to regulate the levels of N-acyl amino acids cooperatively. These data provide evidence for enzymatic division of labor as an enabling biochemical strategy for controlling the levels of a bioactive lipid family.

## Results

### Detection of a second, PM20D1-independent N-acyl amino acid hydrolysis activity

To characterize additional pathways of N-acyl amino acid metabolism in the absence of PM20D1, we analyzed tissue homogenates from wild-type and PM20D1-KO animals for a residual N-acyl amino acid hydrolysis activity. This assay was selected because of the high sensitivity and signal-to-noise ratio that it provides, which enables robust detection of any residual activities that might be present. Two different prototypical N-acyl amino acid substrates, N-arachidonoyl-phenylalanine (C20:4-Phe) and N-arachidonoyl-glycine (C20:4-Gly), were used as substrates. Following incubation with tissue lysates, the hydrolysis of these N-acyl amino acid substrates to free fatty acid product was quantified by liquid chromatography-mass spectrometry (LC-MS, *Figure 1a*). In wild-type mice, robust hydrolysis of C20:4-Phe was observed in eight of the ten tissues tested, with activities in the range of ~0.01 nmol/min/mg (lung) to 1.0 nmol/min/mg (liver). In PM20D1-KO tissues, the hydrolysis of C20:4-Phe was completely abolished (>99% reduction in each tissue), establishing that PM20D1 is the only enzyme responsible for C20:4-Phe hydrolysis activity (*Figure 1b*). The presence of PM20D1 activity in tissue homogenates reflects potential interactions of PM20D1 with the extracellular matrix or with cell surfaces, as has previously been observed with lipoprotein lipase and other secreted enzymes (*Cryer, 1981*). By contrast, using C20:4-Gly as a substrate, both brain and liver from PM20D1-KO mice maintained a robust second hydrolysis activity (*Figure 1c*). The second PM20D1-independent activity accounted for 70% and 11% of the total C20:4-Gly hydrolysis in brain and liver, respectively. In absolute terms, the residual activity in PM20D1-KO liver was higher (0.10 nmol/min/mg) than that observed in the knockout brain tissue (0.03 nmol/min/mg). These data demonstrate the presence of a second, PM20D1-independent hydrolysis activity in brain and liver for C20:4-Gly. That this residual activity is only present for C20:4-Gly but not C20:4-Phe suggested that this second enzyme might exhibit selectivity for regulating subsets of lipid species within the N-acyl amino acid family.

### Molecular identification of fatty acid amide hydrolase (FAAH) as the residual N-acyl amino acid hydrolase

Because liver homogenates exhibited the most robust PM20D1-independent hydrolysis activity, we initiated an effort to identify the enzyme responsible for this activity. We first began with a candidate approach. PM20D1 is one of five members of the mammalian M20 peptidase family, all of which exhibit peptide bond hydrolysis and condensation activity on a variety of small molecule substrates such as N-acetyl amino acids (*Sass et al., 2006*), N-lactoyl amino acids (*Jansen et al., 2015*), and other dipeptides (*Kim et al., 2019*; *Teufel et al., 2003*). However, it was not known whether any of the other mammalian M20 peptidases could also hydrolyze N-fatty acyl amino acids. We therefore recombinantly expressed each of the mammalian M20 peptidases by transient transfection into

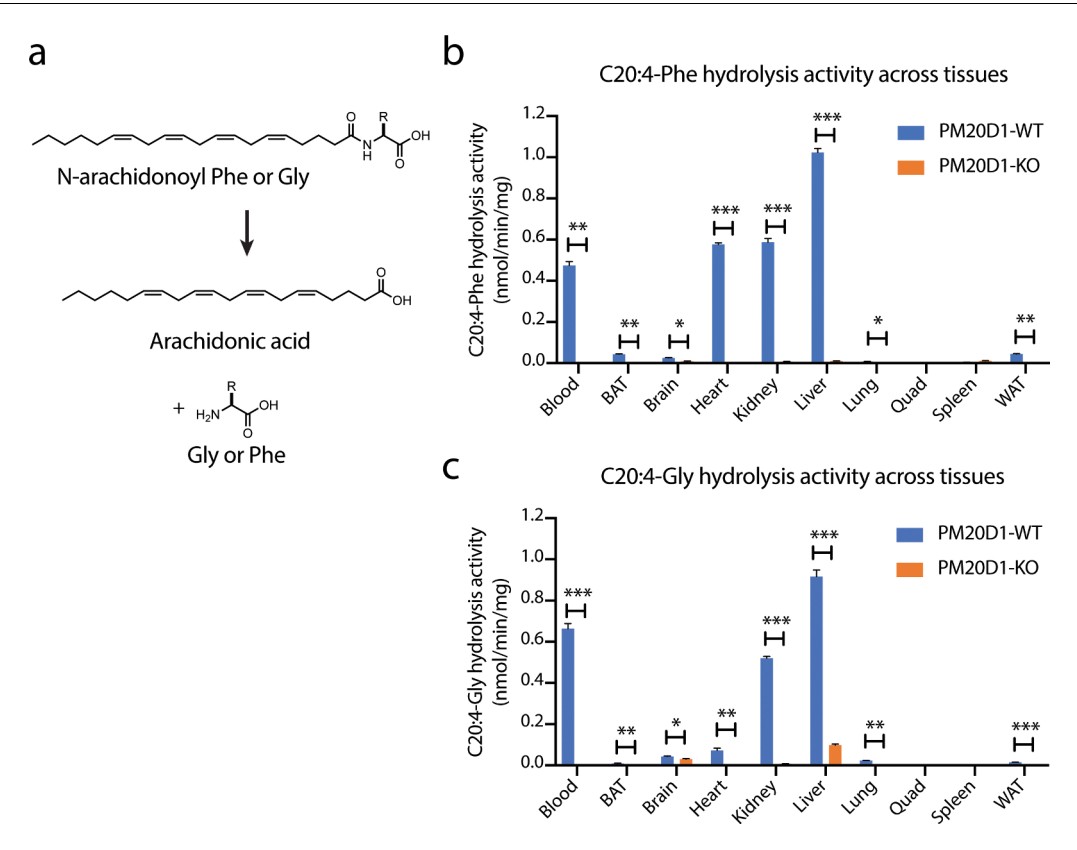

**Figure 1.** Detection of a residual N-acyl amino acid hydrolase activity in PM20D1-KO tissues. (a) Schematic of the enzymatic assay that monitors conversion of C20:4-Phe or C20:4-Gly into arachidonic acid. (b, c) C20:4-Phe (b) and C20:4-Gly (c) hydrolysis activities across the indicated wild-type (blue) or PM20D1-KO (orange) tissues. For (b) and (c), activity assays were conducted with 100 µM substrates and 100 µg tissue lysate in phosphate-buffered saline (PBS) for 1 hr at 37°C. Data are shown as means ± SEM, N = 3/group. All experiments were performed once, with N corresponding to biological replicates. *, p<0.05; **, p<0.01; ***, p<0.001 for the indicated comparison.

HEK293T cells. Cell lysates were harvested and total C20:4-Gly hydrolysis activity was determined by LC-MS. Only PM20D1-transfected cells exhibited robust conversion of C20:4-Gly to arachidonic acid (*Figure 2a*). These data therefore exclude any other mammalian M20 peptidases as a candidate enzyme for catalyzing the residual N-acyl amino acid hydrolysis activity in PM20D1-KO tissues.

We next sought to characterize the enzymological properties of the residual hydrolysis activity from PM20D1-KO livers. Towards this end, livers from PM20D1-KO animals were separated by differential centrifugation at 100,000 x g. The C20:4-Gly hydrolysis activities from the membrane or soluble fraction were measured by LC-MS. Compared to the whole liver, the vast majority of the hydrolysis activity remained in the membrane fraction (*Figure 2b*). This observation suggested that the responsible enzyme contains a transmembrane domain or is otherwise tightly associated with cellular membranes. To determine which enzyme class may be contributing to this activity, we screened a variety of broad-spectrum enzyme inhibitors for their ability to block the C20:4-Gly hydrolysis activity. This panel included the divalent cation chelator EDTA (1 mM), HALT pan-protease inhibitor cocktail (1x), the pan-dipeptidyl peptidase inhibitor talabostat (10 µM), and the serine hydrolase inhibitor MAFP (methyl arachidonoyl fluorophosphonate, 10 µM). Remarkably, pre-treatment of PM20D1-KO liver membrane lysates with MAFP completely abolished the residual C20:4-Gly hydrolysis activity, whereas the other broad-spectrum enzyme inhibitors were entirely without effect (*Figure 2c*). These data strongly suggest that the residual hydrolysis activity is due to a serine hydrolase that is sensitive to MAFP inhibition.

The mammalian serine hydrolases are a large family of more than 250 enzymes (*Bachovchin and Cravatt, 2012*; *Long and Cravatt, 2011*). These hydrolases are characterized by a serine nucleophile that catalyzes hydrolytic reactions via a covalent acyl-enzyme intermediate. We focused on the

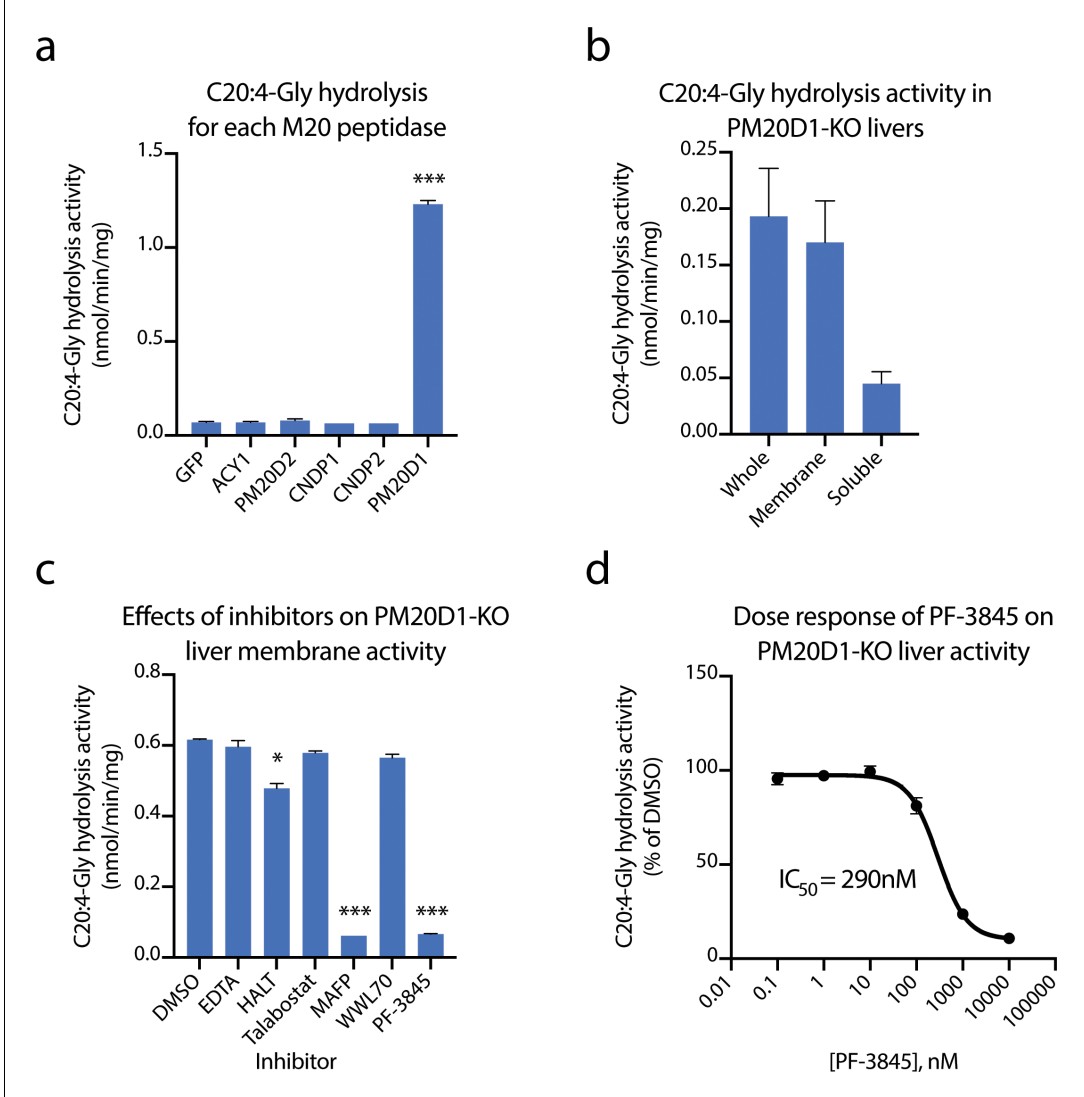

**Figure 2.** Identification of fatty acid amide hydrolase (FAAH) as the enzyme responsible for the PM20D1-independent N-acyl amino acid hydrolase activity. (a, b) C20:4-Gly hydrolysis activity of cell lysates transfected with the indicated mammalian M20 peptidase (a) or of the indicated liver homogenate fraction from PM20D1-KO animals (b). (c, d) Effect on the C20:4-Gly hydrolysis activity from PM20D1-KO liver membranes of the indicated inhibitors. Activity assays were conducted with 100 μM substrates and 100 μg tissue lysate in PBS for 1 hr at 37°C. For panel (b), membrane and soluble fractions of liver lysate were separated by centrifugation at 100,000 x g for 1 hr. For panel (c), inhibitors were pre-incubated at 1 mM for EDTA and 10 μM for all other compounds for 10 min before the start of the assay. Data are shown as means ± SEM, N = 3/group. All experiments were performed once, with N biological replicates. *, p<0.05; ***, p<0.001 for the comparison versus DMSO or GFP control.

subset of approximately 90 enzymes that have been previously described to be covalently inactivated by MAFP (*Bachovchin et al., 2014*). Of these, we excluded the dipeptidyl peptidases (DPPs) because the pan-DPP inhibitor talabostat did not recapitulate the activity of MAFP. From the remaining candidate serine hydrolases, we quickly focused on FAAH. FAAH is best recognized for its role in endocannabinoid signaling via hydrolysis of the cannabinoid receptor agonist anandamide (N-arachidonoyl-ethanolamine) (*Cravatt et al., 1996*; *Cravatt et al., 2001*). Nevertheless, three independent lines of evidence strongly suggest that FAAH may be responsible for the residual N-acyl amino acid hydrolysis activity. First, FAAH contains an N-terminal transmembrane domain and internal intramembrane region, consistent with the observation that the residual hydrolysis activity is localized to liver membranes rather than to the cytosolic fraction (*Patricelli et al., 1998*). Second, FAAH has been shown to regulate several classes of bioactive fatty acid amides, including the N-acyl

ethanolamines and N-acyl taurines, both of which share considerable structural similarity with N-acyl amino acids (*Grevengoed et al., 2019*; *Saghatelian et al., 2004*). Last, FAAH exhibits highest expression in brain and liver in mice, the two anatomical locations where we observe the highest residual N-acyl amino acid hydrolysis activity in PM20D1-KO mice (*Long et al., 2011*).

To determine whether FAAH could contribute to the residual hydrolysis activity in PM20D1-KO livers, we tested the effects of the potent and selective FAAH inhibitor PF-3845 (*Ahn et al., 2009*). This Pfizer compound has previously been shown to be highly selective and to inhibit only FAAH across multiple tissues following administration to mice. Pre-treatment of PM20D1-KO liver membrane lysates with PF-3845 (10 µM) completely blocked the residual hydrolysis activity exactly as was previously observed with MAFP (*Figure 2c*). By contrast, a distinct serine hydrolase inhibitor that does not inhibit FAAH (WWL70, 10 µM) had no effect on the residual C20:4-Gly hydrolysis activity. A dose–response curve established an EC50 of 290 nM for PF-3845, a concentration consistent with the previously reported potency of this compound towards FAAH (*Figure 2d*). Taken together, these data establish FAAH as the enzyme responsible for the residual N-acyl amino acid hydrolysis activity in PM20D1-KO tissues.

## PM20D1 and FAAH exhibit overlapping but distinct substrate specificity in vitro

We next performed alignments of the primary amino acid sequences of mouse PM20D1 and mouse FAAH (*Figure 3a*). As additional comparisons, we also included QRSL1 (glutamyl-tRNA amidotransferase subunit A, mitochondrial), which is the closest murine homolog to FAAH (17% identity), as well as the other four members of the murine M20 peptidase family. PM20D1 was most closely related to ACY1 (24% identity) and shared little identity with FAAH (11%). Our clustering also revealed a closer relationship of PM20D2 with both FAAH and QRSL1 than with the other M20 peptidase family members (*Figure 3a*).

To determine whether N-acyl amino acids are direct FAAH substrates in vitro, we generated recombinant FAAH protein by transient transfection of a C-terminal flag-tagged FAAH construct into HEK293T cells. As a direct comparison, recombinant PM20D1 was generated in parallel. As expected, FAAH was localized entirely intracellularly, consistent with its previously described localization as an ER-associated transmembrane enzyme, whereas PM20D1 protein was largely found in the conditioned media, consistent with its known annotation as a classically secreted enzyme (*Figure 3b*). Using these transfected lysates (for FAAH) and conditioned media (for PM20D1), hydrolysis activity across a diverse panel of 14 N-acyl amino acid substrates was determined by LC-MS. These 14 substrates varied by both amino acid head group and fatty acid chain. FAAH-transfected cells showed robust hydrolysis activity for four N-acyl amino acids tested: C18:1-Gly, C18:1-Ser, C20:4-Gly, and C20:4-Ser (*Figure 3c*). A strong preference was observed for C20:4-Ser over the other three N-acyl amino acid substrates (~1.5 nmol/min/mg for FAAH hydrolysis of C20:4-Ser versus 0.05–0.15 nmol/min/mg for any of the other substrates), at least under these in vitro conditions. In the N-acyl amino acid synthase direction, FAAH also catalyzed the condensation of arachidonic acid with Gly and Ser, but not Phe (*Figure 3d*). By contrast, robust hydrolysis activity was observed for PM20D1 across nearly all members of this N-acyl amino acid substrate panel over mock-transfected background (*Figure 3e*). PM20D1 also efficiently catalyzed the condensation of arachidonic acid with all three Gly, Ser, and Phe amino acids (*Figure 3f*).

To better understand FAAH-mediated N-arachidonoyl glycine hydrolysis activity in the context of its previously described amidase activities, we directly compared the hydrolytic activity of transfected FAAH on C20:4-Gly, anandamide (C20:4-NAE), and N-arachidonoyl-taurine (C20:4-NAT). As expected, FAAH showed robust hydrolysis activity with anandamide as a substrate (0.5 nmol/min/mg) and lower but similar activities with C20:4-NAT and C20:4-Gly (0.04 and 0.02 nmol/min/mg, respectively, *Figure 3—figure supplement 1a*). Conversely, PM20D1 exhibited robust hydrolysis activity only on C20:4-Gly (0.6 nmol/min/mg) and was unable to hydrolyze either anandamide or C20:4-NAT (*Figure 3—figure supplement 1a*). We also determined that C20:4-NAT was not a direct inhibitor for PM20D1 (*Figure 3—figure supplement 1b*). Taken together, these data establish that recombinant FAAH is an N-acyl amino acid synthase/hydrolase in vitro. Our findings also underscore the PM20D1–FAAH pair as an example of convergence in enzymatic activity despite divergence in primary sequence.

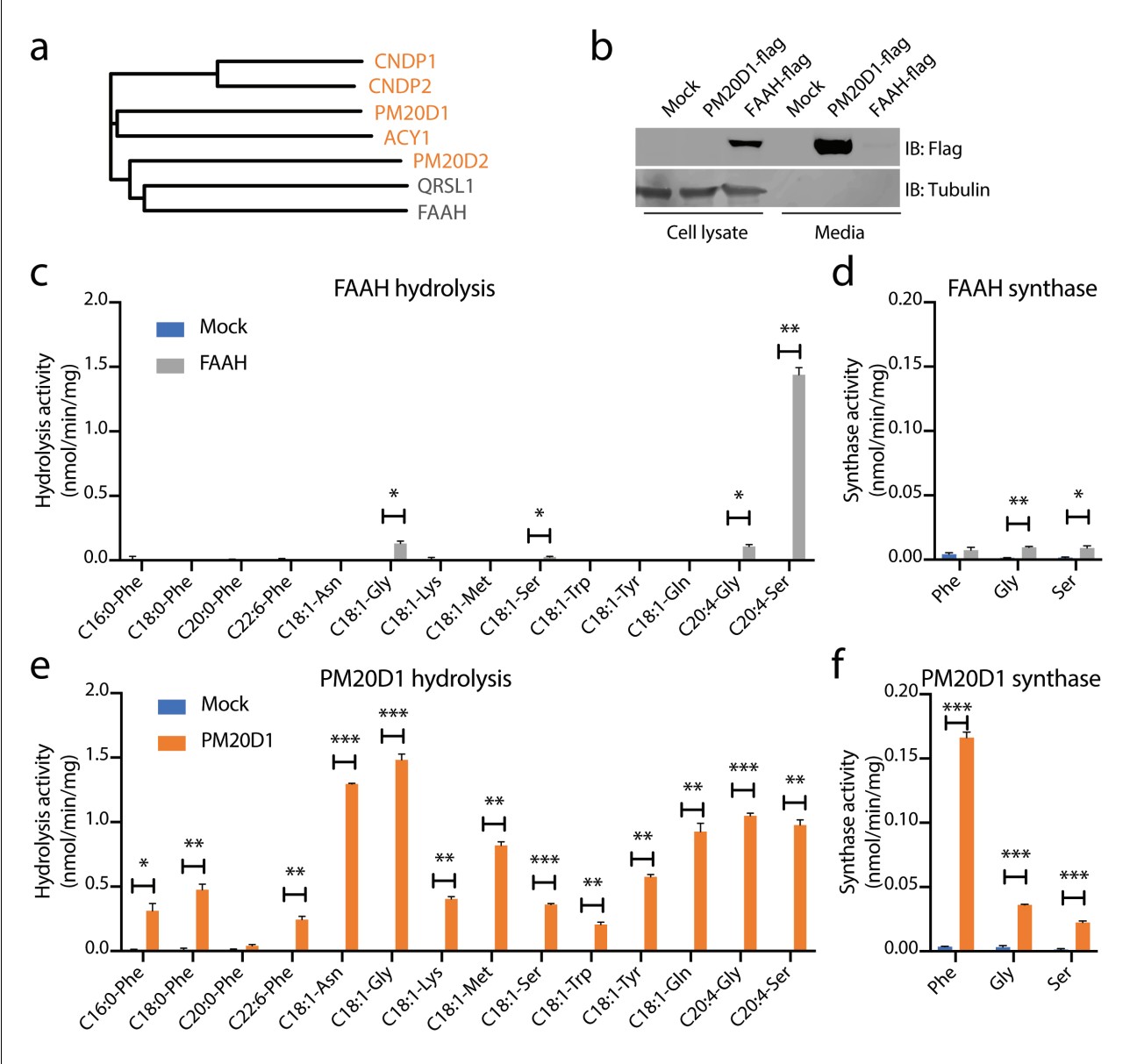

**Figure 3.** N-acyl amino acid hydrolase and synthase substrate scope in vitro for FAAH and PM20D1. (a) Phylogenetic alignment of the five murine M20 peptidases with mouse FAAH and a FAAH-related enzyme, QRSL1. Orange, M20 peptidases; gray, FAAH-related sequences. (b) Anti-flag western blot for cell lysates (left) and conditioned media (right) transfected with the indicated plasmids. (c–f) N-acyl amino acid hydrolysis and synthase activities of FAAH- and mock-transfected cell lysates (b, c) or PM20D1-transfected and mock-transfected conditioned media (d, e). Activity assays were conducted with 100 μM substrates and 100 μg protein in PBS for 1 hr at 37˚C. Data are shown as means ± SEM, N = 3/group. All experiments were performed once, with N biological replicates. *, p<0.05; **, p<0.01; ***, p<0.001 for the indicated comparison.

The online version of this article includes the following figure supplement(s) for figure 3:

**Figure supplement 1.** Additional characterization of FAAH and PM20D1 enzyme activities in vitro.

## N-acyl amino acid metabolism in mice with selective FAAH blockade

Although FAAH is a dominant regulator of several classes of bioactive fatty acid amides in vivo (*Cravatt et al., 2001*; *Grevengoed et al., 2019*; *Han et al., 2013*; *Saghatelian et al., 2004*), the physiologic role of FAAH in regulating N-acyl amino acids has not been systematically explored. Furthermore, considering the intracellular localization of FAAH compared with PM20D1, whether FAAH has a broad role or whether it regulates specific pools of N-acyl amino acids remains unknown. To determine the relevance of the FAAH/N-acyl amino acid pathway in vivo, we measured endogenous

N-acyl amino acid levels following blockade of FAAH in mice. Because both genetic and pharmacological reagents for selective FAAH blockade were available, we performed three independent comparisons: global FAAH-KO versus FAAH-WT mice, a single administration of PF-3845 versus vehicle (10 mg/kg intraperitoneally, 'acute'), or a three-day administration of PF-3845 versus vehicle (10 mg/kg intraperitoneally once per day, 'chronic'). These three comparisons were selected because they had previously been validated to cause dramatic accumulation of other physiologic FAAH substrates in vivo (*Long et al., 2011*).

Liver and blood were harvested and N-acyl amino acids were extracted by homogenization in acetonitrile/methanol. We developed a targeted liquid chromatography-triple quadrupole mass spectrometry (LC-QQQ) method to measure a panel of oleoyl- or arachidonoyl-containing N-acyl amino acids, reasoning that such a set would broadly capture a diverse and representative panel of this lipid family. In these experiments, we were able to detect 26 and 14 N-acyl amino acid species in liver and blood, respectively (*Figure 4a* and *Figure 4—source data 1* and *2*). In the liver, distinct bidirectional changes in N-acyl amino acids were observed in each of the three perturbations. Those changes that were statistically significant across all three conditions of FAAH blockade were elevations in C20:4-Glu (by 2.1-fold) and decreases in C20:4-Gly (by 70%). In other cases, certain N-acyl amino acids changes were only observed in either the genetic (e.g., increased C20:4-Leu/Ile) or pharmacological (e.g., increased C18:1-Glu) model. In the blood, no N-acyl amino acids were significantly changed over controls across all three experiments (*Figure 4b*). We confirmed that PM20D1 activity is not altered in FAAH-KO plasma (*Figure 4—figure supplement 1*). By abundance, hepatic N-acyl amino acids levels were similar to N-acyl ethanolamine and N-acyl taurines in wild-type mice (*Figure 4—source data 1* and *2*). Taken together, these data demonstrate that FAAH is a bidirectional regulator of a subset of intracellular, but not extracellular, N-acyl amino acids.

## Cooperative regulation of N-acyl amino acids by PM20D1 and FAAH in vivo

Our data establish that at least two enzymes, PM20D1 and FAAH, contribute to the regulation of endogenous N-acyl amino acid levels. Individual blockade of PM20D1 or FAAH resulted in bidirectional dysregulation of N-acyl amino acids. We therefore considered the possibility that dual inhibition of both PM20D1 and FAAH would result in a complete ablation of N-acyl amino acid synthase/hydrolase activities, and in concomitant global elevations or global depletions of endogenous N-acyl amino acid levels. To test this hypothesis, we used global PM20D1-KO animals in combination with a FAAH inhibitor to block both PM20D1 and FAAH simultaneously. PM20D1-KO animals were chronically treated with PF-3845 (10 mg/kg intraperitoneally once per day for three days). As controls, PM20D1-KO or PM20D1-WT littermates were administered vehicle control (saline) in parallel. First, we measured liver N-acyl amino acid hydrolysis activity from each of the three groups of mice. As we described previously, livers from vehicle-treated PM20D1-KO mice exhibited a residual C20:4-Gly hydrolysis activity when compared to livers from PM20D1-WT mice. Following PF-3845 treatment in PM20D1-KO mice, the residual hepatic C20:4-Gly hydrolysis activity was entirely abolished (*Figure 5a*). These data establish that PM20D1 and FAAH are the only two C20:4-Gly hydrolysis activities in liver, at least under the assay conditions used here, and further validate our previous in vitro studies (*Figure 2c,d*).

Next, we measured endogenous N-acyl amino acid levels in both liver and blood. Under basal conditions, PM20D1-KO mice exhibit bidirectional dysregulation of several N-acyl amino acids compared to PM20D1-WT mice (*Figure 5b* and *Figure 5—source data 1*). Surprisingly, complete ablation of N-acyl amino acid synthase/hydrolase activities did not uniformly change N-acyl amino acids in a positive or negative direction. Instead, dual inhibition of PM20D1 and FAAH uncovered a remarkable cooperativity of these two enzymatic pathways in the regulation of N-acyl amino acids. In general, individual regulation by PM20D1 or FAAH were not predictive of those N-acyl amino acid species that were regulated by both enzymatic pathways together or even the directionality of change. For instance, hepatic N-acyl serines were largely unaltered by individual blockade of either FAAH or PM20D1 alone (*Figure 4a* and *Figure 5b*). Following dual blockade of both enzymes, C18:1-Ser and C20:4-Ser were dramatically accumulated by 13-fold and 26-fold, respectively (*Figure 5b*). In other cases, dual blockade resulted in changes to N-acyl amino acids in the opposite direction compared to individual blockade alone. For instance, C18:1-Gly was reduced in PM20D1-KO livers and unaltered by FAAH blockade, but nevertheless accumulated to 8-fold over controls

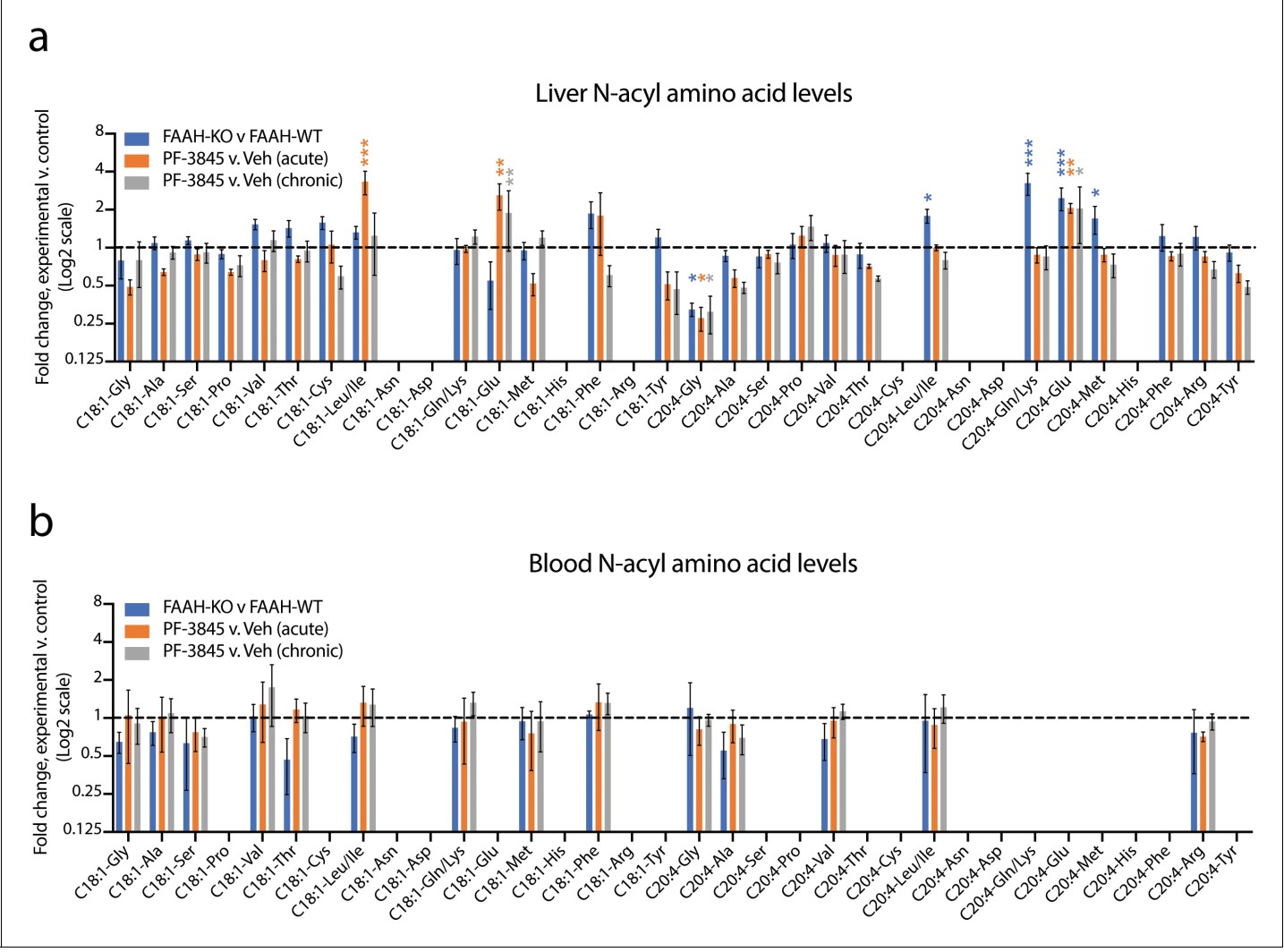

**Figure 4.** Changes in N-acyl amino acids upon selective blockade of FAAH in vivo. (**a, b**) Fold change of the indicated N-acyl amino acids compared to the control for each of the indicated comparisons from liver (**a**) or blood (**b**). For drug treatment, PF-3845 was administered intraperitoneally at 10 mg/kg once (acute) or for three consecutive days (chronic). Tissues were harvested 3 hr after the final dose. No bars are shown for N-acyl amino acids that were below the limit of detection. Data are shown as means ± SEM, N = 4–5 mice/group for each of the indicated comparisons. All experiments were performed once, with N biological replicates. *, p<0.05; **, p<0.01; ***, p<0.001 by ANOVA with Dunnett's multiple comparisons test versus control animals.

The online version of this article includes the following source data and figure supplement(s) for figure 4:

**Source data 1.** Absolute quantitation of N-acyl amino acids in liver and plasma following FAAH blockade.

**Source data 2.** Absolute quantitation of N-acyl ethanolamines and N-acyl taurines in wild-type mouse liver.

**Figure supplement 1.** PM20D1 activity in FAAH-KO plasma.

when both enzymes were inhibited. In the blood, similar cooperative interactions could also be observed (*Figure 5c*). In some cases, the directionality of the N-acyl amino acid changes was concordant in blood and liver (e.g., N-acyl glycines and N-acyl leucines), whereas in other cases (e.g., C18:1-Met), the intracellular pools accumulated with a concomitant decrease in extracellular levels. These data establish that PM20D1 and FAAH engage in non-additive crosstalk in vivo to regulate intracellular and circulating levels of N-acyl amino acids .

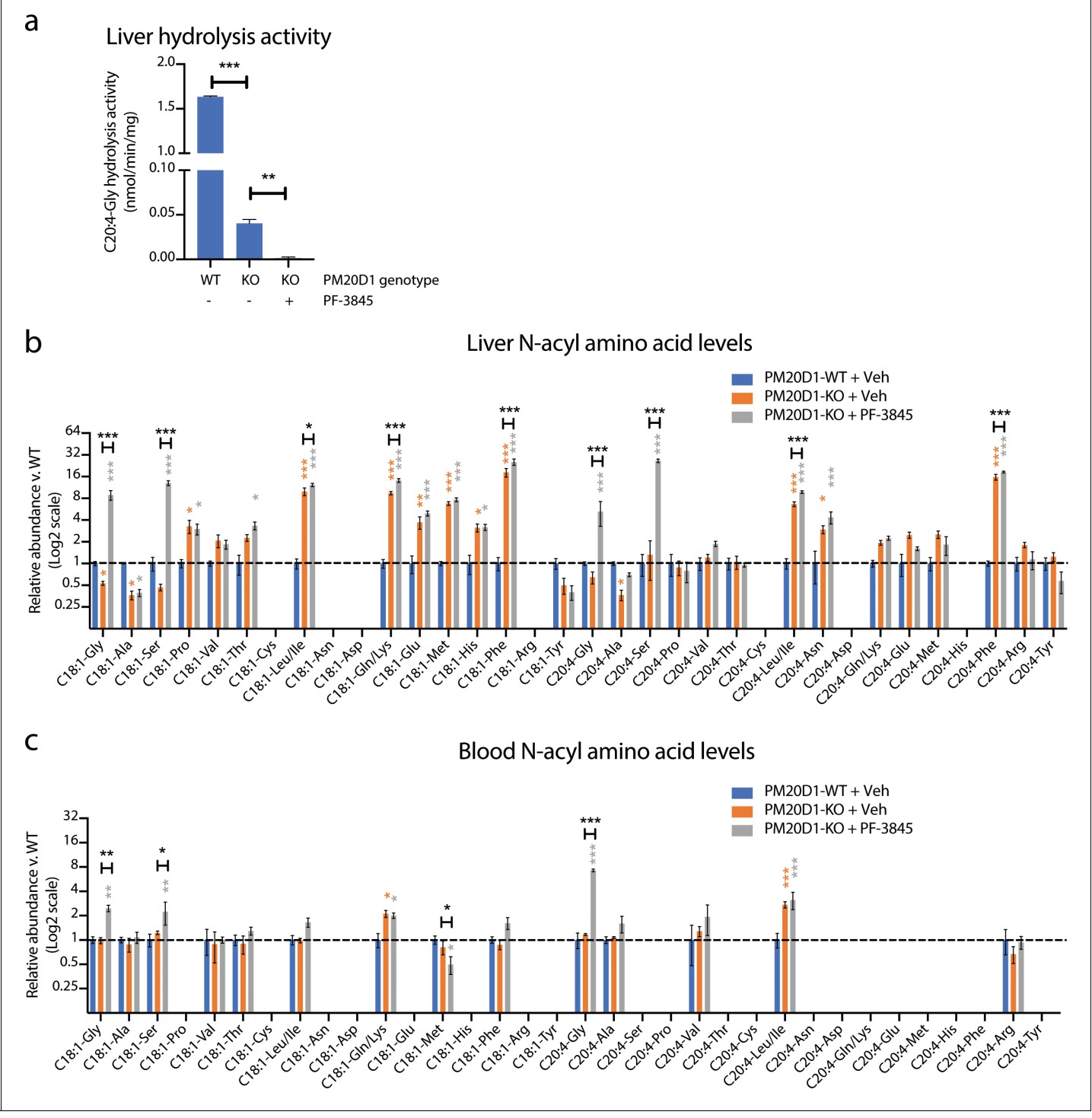

**Figure 5.** Cooperative interactions between PM20D1 and FAAH regulate endogenous N-acyl amino acid levels. (a) C20:4-Gly hydrolysis activity in livers from PM20D1-WT, PM20D1-KO, or PM20D1-KO treated with PF-3845. (b, c) Relative fold change of the indicated N-acyl amino acids in PM20D1-KO mice or in PM20D1-KO mice treated with PF-3845 versus wild-type mice in liver (b) or blood (c). For drug treatment, PF-3845 was administered intraperitoneally at 10 mg/kg for three consecutive days and tissues were harvested 3 hr after the final dose. No bars are shown for N-acyl amino acids that were below the limit of detection. Data are shown as means ± SEM, N = 4–5 mice/group for each of the indicated comparisons. All experiments were performed once, with N biological replicates. *, p<0.05; **, p<0.01; ***, p<0.001 in color are versus PM20D1-WT levels, whereas those in black are for the indicated comparison by ANOVA with Tukey's multiple comparison test.

The online version of this article includes the following source data for figure 5:

**Source data 1.** Absolute quantitation of N-acyl amino acids in liver and plasma following PM20D1 or dual PM20D1/FAAH blockade.

## Discussion

The N-acyl amino acids are a diverse bioactive lipid family. Using PM20D1-KO tissues as a discovery tool, we establish FAAH as a second intracellular mammalian N-acyl amino acid synthase/hydrolase. In vitro, FAAH catalyzes the bidirectional synthesis and hydrolysis of a subset of N-acyl amino acids that have a narrower substrate specificity than PM20D1. Genetic ablation or pharmacological inhibition of FAAH in vivo established that this enzyme is also a physiological regulator of intracellular but not of extracellular N-acyl amino acids. Our data uncover the PM20D1–FAAH pair as an example of enzymatic convergence, despite largely unrelated primary amino acid sequences. More generally, these findings underscore enzymatic and spatial division of labor as a mechanism for the control of subsets of a diverse thermogenic lipid family.

The identification of two mammalian N-acyl amino acid synthase/hydrolase enzymes, one localized extracellularly and one intracellularly, raises important questions about the similarities and differences between the N-acyl amino acids within each of these localizations and about crosstalk between the intracellular and extracellular pools of N-acyl amino acids. What factors determine the compartmentalization of N-acyl amino acids? What are the mechanisms by which these lipids are imported into or exported out of cells? What functional differences are there between these two pools of N-acyl amino acids? One possibility is that certain N-acyl amino acid bioactivities require interactions within the cell (e.g., binding to mitochondria to stimulate respiratory uncoupling) whereas others involve extracellular interactions (e.g., engaging cell surface receptors), suggesting that certain compartmentalized pools of N-acyl amino acids might be more relevant for specific bioactivities. Identifying specific transporters for N-acyl amino acids may help to clarify the relative contribution of the intracellular versus extracellular pools of these lipids. The many examples of intra- and inter-cellular transport of sterols (*Phillips, 2014*), fatty acids (*Pepino et al., 2014*), and phospholipids (*Hankins et al., 2015*) provide a fertile starting point for discovering analogous pathways for the transport and compartmentalization of N-acyl amino acids.

FAAH has also been implicated in diverse physiologic conditions in mice and humans. Pharmacological or genetic blockade of FAAH has been shown to regulate pain and inflammation (*Ahn et al., 2009*; *Cravatt et al., 2001*), obesity and metabolic homeostasis (*Brown et al., 2012*; *Touriño et al., 2010*), and anxiety and depression (*Gobbi et al., 2005*; *Kathuria et al., 2003*), amongst many others. Polymorphisms in the *FAAH* gene have also provided human genetic evidence for these disease associations (*Sipe et al., 2002*; *Sipe et al., 2005*). These phenotypes are classically associated with anandamide elevation and activation of the cannabinoid receptors upon FAAH inhibition or genetic deletion. However, beyond the endocannabinoid system, FAAH also regulates several other classes of bioactive lipids including other N-acyl ethanolamines, N-acyl taurines, and now, N-acyl amino acids. We propose that FAAH-regulated N-acyl amino acids may also contribute to some of these observed phenotypes. Projecting forward, critical tests of the FAAH/N-acyl amino acid contribution without confounding contributions from other FAAH-regulated lipids will require the identification of selective FAAH point mutants that only catalyze the synthesis or hydrolysis of specific species of N-acyl amino acids, but not other fatty acid amide substrates.

Our data reveal that complete ablation of all C20:4-Gly synthase/hydrolase activities via dual blockade of both PM20D1 and FAAH was not sufficient to elevate or deplete all endogenous N-acyl amino acids globally. Instead, we observed extensive and non-additive interactions between these two enzymatic pathways in the regulation of specific subsets of N-acyl amino acids. In general, these non-additive interactions could not be predicted by in vitro substrate specificity or even individual N-acyl amino acid regulation. To the best of our knowledge, this type of cooperativity has not been previously described for any biochemical pathway. Quantitative flux analysis for the various amino acid, fatty acid, and N-acyl amino acid components will be required to understand how these metabolic fluxes are re-wired upon blockade of each enzyme individually or together.

Last, our findings suggest that additional biochemical pathways beyond PM20D1 and FAAH contribute to regulating the endogenous levels of this lipid family. Potential candidate pathways include additional amidase enzymes on other N-acyl amino acid substrates, enzymes that catalyze the conjugation of fatty acid CoAs with amino acids, or non-mammalian sources (*Brady et al., 2004*; *Cohen et al., 2017*; *Jeffries et al., 2016*; *Van Wagoner and Clardy, 2006*). Molecular identification of these additional pathways of N-acyl amino acid metabolism will ultimately enable the dissection

and therapeutic manipulation of more specific subsets of this diverse bioactive lipid family in organismal physiology.

# Materials and methods

**Key resources table**

| Reagent type (species) or resource | Designation | Source or reference | Identifiers | Additional information |
|---|---|---|---|---|
| Mouse line (*Mus musculus*) | PM20D1-KO | *Long et al., 2018* (PMID:29967167) | | |
| Mouse line (*M. musculus*) | C57BL/6J | Jackson Labs | 000664 | |
| Transfected construct (*M. musculus*) | PM20D1-flag | Addgene | 84566 | |
| Transfected construct (*M. musculus*) | FAAH-flag | Origene | MR209084 | |
| Transfected construct (*M. musculus*) | ACY1-flag | Origene | MR206415 | |
| Transfected construct (*M. musculus*) | CNDP1-flag | Origene | MR219018 | |
| Transfected construct (*M. musculus*) | CNDP2-flag | Origene | MR207616 | |
| Transfected construct (*M. musculus*) | PM20D2-flag | Origene | MR222068 | |
| Cell line (*Homo sapiens*) | HEK293T | ATCC | CRL-3216 | |
| Antibody | Anti-flag M2, mouse monoclonal | Sigma | F1804 | (1:1000) |
| Antibody | Anti-tubulin, rabbit polyclonal | Abcam | Ab6046 | (1:1000) |
| Chemical compound | PF-3845 | Selleckchem | S2666 | |
| Chemical compound | C20:4-Gly | Cayman | 90051 | |
| Chemical compound | C20:4-Ser | Cayman | 10005455 | |
| Chemical compound | C20:4-Phe | Abcam | Ab141612 | |
| Chemical compound | Arachidonic acid | Sigma-Aldrich | 10931 | |
| Chemical compound | WWL70 | Sigma-Aldrich | SML1641 | |
| Chemical compound | Talabostat | R and D | 3719 | |
| Chemical compound | MAFP | Fisher Scientific | 14-21-5 | |
| Chemical compound | C20:4-NAT | Cayman | 10005537 | |
| Chemical compound | Anandamide | Sigma-Aldrich | A0580 | |

*Continued on next page*

*Continued*

| Reagent type (species) or resource | Designation | Source or reference | Identifiers | Additional information |
|---|---|---|---|---|
| Chemical compound | C16:0-Phe | *Lin et al., 2018* (PMID:29533650) | | |
| Chemical compound | C18:0-Phe | *Lin et al., 2018* (PMID:29533650) | | |
| Chemical compound | C20:0-Phe | *Lin et al., 2018* (PMID:29533650) | | |
| Chemical compound | C22:6-Phe | *Lin et al., 2018* (PMID:29533650) | | |
| Chemical compound | C18:1-Asn | *Lin et al., 2018* (PMID:29533650) | | |
| Chemical compound | C18:1-Gly | Cayman | 90269 | |
| Chemical compound | C18:1-Lys | *Lin et al., 2018* (PMID:29533650) | | |
| Chemical compound | C18:1-Met | *Lin et al., 2018* (PMID:29533650) | | |
| Chemical compound | C18:1-Ser | Cayman | 13058 | |
| Chemical compound | C18:1-Trp | *Lin et al., 2018* (PMID:29533650) | | |
| Chemical compound | C18:1-Tyr | *Lin et al., 2018* (PMID:29533650) | | |
| Chemical compound | C18:1-Gln | *Lin et al., 2018* (PMID:29533650) | | |

## General animal information

Animal experiments were performed according to procedures approved by the Stanford University IACUC. Mice were maintained in 12 hr light-dark cycles at 22°C and fed a standard irradiated rodent chow diet. All experiments on wild-type mice were performed with male C57BL/6J mice purchased from Jackson Laboratories (stock number 000664). Global *Pm20d1* knockout mice were obtained from Bruce M. Spiegelman (Dana-Farber Cancer Institute) and are available from Jackson Laboratories (stock number 032193). PF-3845 was administered to mice in a solution of 18:1:1 saline:kolliphor EL:DMSO in a volume of 200 µl/mouse (intraperitoneally).

## Materials

N-arachidonoyl glycine, N-arachidonoyl serine, and N-arachidonoyl-taurine were purchased from Cayman. N-arachidonoyl phenylalanine was purchased from Abcam. Arachidonic acid, anandamide, and WWL70 were purchased from Sigma-Aldrich. PF-3845 was purchased from Selleckchem. MAFP and EDTA were purchased from Fisher. Talabostat was purchased from R and D. Non-commercially available N-acyl amino acids were synthesized as previously described (*Lin et al., 2018*; *Long et al., 2016*). Plasmids were obtained from the following sources: mouse PM20D1-flag (Addgene 132682), mouse FAAH-flag (Origene MR209084), mouse ACY1-flag (Origene MR206415), mouse CNDP1-flag (Origene MR219018), mouse CNDP2-flag (Origene MR207616), and mouse PM20D2-flag (MR222068).

## Statistics, sample size estimation, and replicates

All statistical comparisons were performed using Student's t-test or ANOVA with Tukey or Dunnett's multiple comparison test. No explicit power analysis was used to determine sample sizes. Sample sizes were determined on the basis of previous literature for biochemical or animal studies. All experiments were performed once, with N corresponding to biological replicates. Outliers were not removed from analyses. The experimentalist was not blinded to sample or treatment conditions.

## Cell culture

HEK293T cells were obtained from ATCC (CRL-3216) and cultured in DMEM with L-glutamine, 4.5 g/L glucose and sodium pyruvate (Corning 10013CV) supplemented with 10% FBS (Corning 35010CV). Cells were incubated at 37°C in 5% $CO_2$ for growth and tranfections. All cell lines were authenticated by DNA fingerprint STR analysis by ATCC. Mycoplasma was not tested. Authentication of cell lines beyond ATCC was not completed due to laboratory disruptions by COVID-19.

## Production of recombinant enzymes

Plasmids were transiently transfected into HEK293T cells using PolyFect (Qiagen) according to the manufacturer's instructions. The medium was changed to serum-free DMEM one day post-transfection. After an additional 24 hr, the medium was collected and the cells were harvested by scraping.

## Molecular studies

Western blotting was performed according to standard methods. The following antibodies were used: anti-flag M2 antibody (Sigma F1804, diluted 1:10,000), and tubulin (Abcam ab6046, diluted 1:10,000).

## Enzyme activity assays in vitro

Plasma was collected from mice and used directly for the activity assays. Tissues were homogenized using a Benchmark BeadBlaster Homogenizer in ice-cold PBS, centrifuged to remove debris (5 min at 1000 x g), and the supernatant was collected and used for activity assays. For assays using liver membranes, total liver homogenates were transferred into ultracentrifuge inserts and spun at 100,000 x g on a Beckman Centrifuge I8-70M for 1 hr at 4°C. In vitro enzymatic reactions were conducted in glass vials and initiated by the addition of 100 µg protein. Final reaction conditions for the hydrolase reactions were 100 µM substrate (C20:4-Gly or C20:4-Phe) and 100 µg protein in 100 µl PBS, and for the synthase reactions were 1 mM Phe, 1 mM oleic acid, and 100 µg protein in 100 µl PBS. After 1 hr at 37°C, reactions were quenched with 400 µl 2:1 v/v acetonitrile:methanol and vortexed. Reaction vials were centrifuged at 2000 x g to remove debris, and the supernatant was collected and analyzed by LC-MS as described below. For inhibitor assays, tissue lysates were treated with the indicated inhibitors for 10 min at room temperature before the introduction of the indicated substrates.

## Extraction of N-acyl amino acids from blood and tissues

Frozen plasma (30 µl) were extracted in 160 µl of 1:1 v/v acetonitrile:methanol. Liver tissues were extracted in 500 µl 2:2:1 v/v/v acetonitrile:methanol:water on a BeadBlaster homogenizer for 1 min. Extracts were centrifuged (10 min, 5000 x g) to remove debris. The supernatant was isolated and centrifuged again (10 min, 5000 x g). Finally, the twice-clarified supernatant was transferred to a mass spectrometry vial and analyzed by LC-MS as described below.

## Measurements of N-acyl amino acids in vivo and enzyme activities in vitro by LC-MS

Mass spectrometry analysis was performed with an electrospray ionization (ESI) source on an Agilent 6470 Triple Quadrupole (QQQ). For separation of metabolites, normal-phase chromatography was performed with a Luna 5 µm C5 100 Å LC column (Phenomenex 00B-4043-E0). The mobile phases were as follows: Buffer A, 95:5 water/methanol; Buffer B, 60:35:5 isopropanol/methanol/water with 0.1% ammonium hydroxide in both Buffer A and B for negative ionization mode. For AJS ESI ion source parameters, the drying gas temperature was set to 250°C with a flow rate of 12 l/min, and the nebulizer pressure was 25 psi. The sheath gas temperature was set to 300°C with a flow rate of 12 l/min. The capillary voltage was set to 2500 V and the fragmentor voltage was set to 135 V. For the measurement of in vitro enzyme activity assays, the flow rate for each run started at 95% A/5% B for 3 min at 0.6 ml/min, followed by a gradient starting at 95% A/5% B changing linearly to 5% A/95% B over the course of 3 min at 0.6 ml/min, followed by 5% A/95% B for 1.5 min at 0.6 ml/min. For the measurement of metabolites from blood and liver in vivo, the flow rate for each run started at 95% A/5% B for 1 min at 0.6 ml/min, followed by a gradient starting at 95% A/5% B changing

linearly to 5% A/95% B over the course of 10 min at 0.6 ml/min, followed by 5% A/95% B for 3 min at 0.6 ml/min, and back to 95% A/5% B over 1 min at 0.6 ml/min.

The QQQ acquisition parameters were as follows. For in vitro assays, the mass range was set from 100 to 500 m/z. For measurement of endogenous N-acyl amino acids, metabolites were detected by the SRM of the transition from precursor to product ions (corresponding to the amino acid fragment) at collision energy 20. The following table includes the list of transitions used. N-acyl taurines and N-acyl ethanolamines were measured as described previously (*Long et al., 2011*).

| Compound name | Precursor ion | Product ion | Dwell | Fragmentor | Collision energy | Cell accelerator voltage | Polarity |
|---|---|---|---|---|---|---|---|
| C18:1-Trp | 467.3 | 203.1 | 50 | 135 | 20 | 5 | Negative |
| C20:4-Tyr | 466.3 | 180.1 | 50 | 135 | 20 | 5 | Negative |
| C20:4-Arg | 459.4 | 173.1 | 50 | 135 | 20 | 5 | Negative |
| C20:4-Phe | 450.3 | 164.1 | 50 | 135 | 20 | 5 | Negative |
| C18:1-Tyr | 444.3 | 180.1 | 50 | 135 | 20 | 5 | Negative |
| C20:4-His | 440.3 | 154.1 | 50 | 135 | 20 | 5 | Negative |
| C18:1-Arg | 437.4 | 173.1 | 50 | 135 | 20 | 5 | Negative |
| C20:4-Met | 434.3 | 148 | 50 | 135 | 20 | 5 | Negative |
| C20:4-Glu | 432.3 | 146.1 | 50 | 135 | 20 | 5 | Negative |
| C20:4-Gln/Lys | 431.3 | 145.1 | 50 | 135 | 20 | 5 | Negative |
| C18:1-Phe | 428.3 | 164.1 | 50 | 135 | 20 | 5 | Negative |
| C18:1-His | 418.3 | 154.1 | 50 | 135 | 20 | 5 | Negative |
| C20:4-Asp | 418.3 | 132 | 50 | 135 | 20 | 5 | Negative |
| C20:4-Asn | 417.3 | 131.1 | 50 | 135 | 20 | 5 | Negative |
| C20:4-Leu/Ile | 416.4 | 130.1 | 50 | 135 | 20 | 5 | Negative |
| C18:1-Met | 412.3 | 148 | 50 | 135 | 20 | 5 | Negative |
| C18:1-Glu | 410.3 | 146.1 | 50 | 135 | 20 | 5 | Negative |
| C18:1-Gln/lys | 409.3 | 145.1 | 50 | 135 | 20 | 5 | Negative |
| C20:4-Cys | 406.3 | 120 | 50 | 135 | 20 | 5 | Negative |
| C20:4-Thr | 404.3 | 118.1 | 50 | 135 | 20 | 5 | Negative |
| C20:4-Val | 402.3 | 116.1 | 50 | 135 | 20 | 5 | Negative |
| C20:4-Pro | 400.3 | 114.1 | 50 | 135 | 20 | 5 | Negative |
| C18:1-Asp | 396.3 | 132 | 50 | 135 | 20 | 5 | Negative |
| C18:1-Asn | 395.3 | 131.1 | 50 | 135 | 20 | 5 | Negative |
| C18:1-Leu/Ile | 394.4 | 130.1 | 50 | 135 | 20 | 5 | Negative |
| C20:4-Ser | 390.3 | 104 | 50 | 135 | 20 | 5 | Negative |
| C15-Phe | 388.3 | 164.1 | 50 | 135 | 20 | 5 | Negative |
| C18:1-Cys | 384.3 | 120 | 50 | 135 | 20 | 5 | Negative |
| C18:1-Thr | 382.3 | 118.1 | 50 | 135 | 20 | 5 | Negative |
| C18:1-Val | 380.3 | 116.1 | 50 | 135 | 20 | 5 | Negative |
| C18:1-Pro | 378.3 | 114.1 | 50 | 135 | 20 | 5 | Negative |
| C20:4-Ala | 374.3 | 88 | 50 | 135 | 20 | 5 | Negative |
| C18:1-Ser | 368.3 | 104 | 50 | 135 | 20 | 5 | Negative |
| C20:4-Gly | 360.3 | 74 | 50 | 135 | 20 | 5 | Negative |
| C18:1-Ala | 352.3 | 88 | 50 | 135 | 20 | 5 | Negative |
| C18:1-Gly | 338.3 | 74 | 50 | 135 | 20 | 5 | Negative |
| C20:4-Trp | 489.3 | 203.1 | 50 | 135 | 20 | 5 | Negative |

## Acknowledgements

We thank members of the Long and Svensson labs for helpful discussions and K Masuda and B F Cravatt (Scripps Research) for tissues from the FAAH-KO mice. This work was supported by the US National Institutes of Health (DK105203 and DK124265 to JZL), by the Stanford ChEM-H Institute (CRF), and by the Stanford Diabetes Research Center (P30DK116074).

## Additional information

### Funding

| Funder | Grant reference number | Author |
| --- | --- | --- |
| National Institutes of Health | DK105203 | Jonathan Long |
| Stanford Diabetes Research Center | P30DK116074 | Jonathan Z Long |
| National Institutes of Health | DK124265 | Jonathan Z Long |
| Stanford University | Stanford ChEM-H Institute | Curt R Fischer |

The funders had no role in study design, data collection and interpretation, or the decision to submit the work for publication.

### Author contributions

Joon T Kim, Conceptualization, Formal analysis, Investigation, Methodology; Stephanie M Terrell, Veronica L Li, Wei Wei, Investigation; Curt R Fischer, Investigation, Methodology; Jonathan Z Long, Conceptualization, Data curation, Formal analysis, Supervision, Funding acquisition, Visualization, Writing - original draft, Writing - review and editing

### Author ORCIDs

Joon T Kim https://orcid.org/0000-0002-8663-1414
Curt R Fischer http://orcid.org/0000-0003-3386-9813
Jonathan Z Long https://orcid.org/0000-0003-2631-7463

### Ethics

Animal experimentation: This study was performed in strict accordance with the recommendations in the Guide for the Care and Use of Laboratory Animals of the National Institutes of Health. Animal experiments were performed according to procedures approved by the Stanford University IACUC, protocol APLAC-32841.

### Decision letter and Author response

Decision letter https://doi.org/10.7554/eLife.55211.sa1
Author response https://doi.org/10.7554/eLife.55211.sa2

## Additional files

### Supplementary files

- Transparent reporting form

### Data availability

All data generated or analysed during this study are included in the manuscript.

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
