## [Decision Letter]

**Acceptance summary:**

N-acyl amino acids are an emerging family of lipid signaling molecules composed of a fatty acid chain conjugated to amino acids. These bioactive lipids have potentially far-reaching impacts on metabolism, yet their regulation is still not fully understood. This paper identifies fatty acid amide hydrolase (FAAH) as an enzyme that acts in concert with a previously identified enzyme PM20D1 to regulate levels of the N-acyl amino acids. Identification of this additional pathway will enable further dissection of the metabolic roles of this bioactive lipid family.

**Decision letter after peer review:**

Thank you for submitting your article "Cooperative enzymatic control of N-acyl amino acids by PM20D1 and FAAH" for consideration by *eLife*. Your article has been reviewed by Michael Marletta as the Senior Editor, a Reviewing Editor, and three reviewers. The following individuals involved in review of your submission have agreed to reveal their identity: Matthew Paul Gillum (Reviewer #2).

The reviewers have discussed the reviews with one another and the Reviewing Editor has drafted this decision to help you prepare a revised submission.

Summary:

Kim et al. report the identification of fatty acid amide hydrolase (FAAH) as an enzyme that regulates levels of N-acyl amino acids together with previously identified PM20D1. The N-acyl amino acids are an emerging family of lipid signals with potentially far-reaching impacts on metabolism, and this work demonstrates how FAAH and PM20D1 act in concert to regulate the levels of this signal. All three reviewers were supportive and agreed that this work was an interesting contribution to this field. A few concerns were raised that the authors should be able to address with additional data and/or clarifying explanations.

Essential revisions:

1) A major question is whether FAAH's primary substrate in the liver N-acyl taurines could be indirectly causing the effects observed in vivo. First, it would be good to know the concentrations of N-acyl amino acids compare to the NATs (or NAEs in the brain). Most of the data in the manuscript are reported as a fold-change, but it would be good to know the relative levels of FAAH substrates in vivo to know if the N-acyl amino acids constitute a significant substrate source for FAAH. Second, a likely explanation is that the absence of FAAH regulates some N-acyl amino acids because these are direct FAAH substrates. However, another model that might lead to similar data is that the absence of FAAH results in high NAT levels that can regulate PM20D1 leading to increased levels of some N-acyl amino acids. What are the effects on PM20D1 expression in FAAH-KO mice? If there are effects, it would be good to check that NATs are not secondary PM20D1 substrates or inhibitors that could influence the activity of this enzyme. The result would be interesting regardless since it could reveal additional crosstalk between the systems. In vitro assays with tissue lysates and pure enzymes +/- NATs should suffice.

2) As a related point to #1, it would be useful to discuss (or measure) the rate of FAAH-mediated N-acyl taurine hydrolysis compared to that for N-acyl amino acid hydrolysis, to put this new FAAH activity in context.

3) The activity of PM20D1 appears high in tissue (i.e. liver and kidney), though it is discussed as a secreted enzyme? Can the authors clarify this?

---

## [Author Response]

Essential revisions:1) A major question is whether FAAH's primary substrate in the liver N-acyl taurines could be indirectly causing the effects observed in vivo. First, it would be good to know the concentrations of N-acyl amino acids compare to the NATs (or NAEs in the brain). Most of the data in the manuscript are reported as a fold-change, but it would be good to know the relative levels of FAAH substrates in vivo to know if the N-acyl amino acids constitute a significant substrate source for FAAH.

We thank the reviewers for these suggestions. We have now provided absolute quantitation of hepatic N-acyl amino acids, N-acyl ethanolamines, and N-acyl taurines in three new source data files (Figure 4—source data 1, Figure 4—source data 2, and Figure 5—source data 1). These data show that N-acyl amino acids are in the range of 10-800 pmol/g, depending on the precise species and experiment. For comparison, N-acyl taurines were on average ~400 pmol/g as were N-acyl ethanolamines (average ~100 pmol/g). In the main text, we have now added the following sentence to subsection “N-acyl amino acid metabolism in mice with selective FAAH blockade”: “By abundance, hepatic N-acyl amino acids levels were similar to N-acyl ethanolamine and N-acyl taurines in wild-type mice (Figure 4—source data 1 and Figure 4—source data 2).”

Second, a likely explanation is that the absence of FAAH regulates some N-acyl amino acids because these are direct FAAH substrates. However, another model that might lead to similar data is that the absence of FAAH results in high NAT levels that can regulate PM20D1 leading to increased levels of some N-acyl amino acids. What are the effects on PM20D1 expression in FAAH-KO mice? If there are effects, it would be good to check that NATs are not secondary PM20D1 substrates or inhibitors that could influence the activity of this enzyme. The result would be interesting regardless since it could reveal additional crosstalk between the systems. In vitro assays with tissue lysates and pure enzymes +/- NATs should suffice.

To directly address this question, we have measured PM20D1 activity in plasma from WT and FAAH-KO mice. As shown in Figure 4—figure supplement 1, we did not observe any differences in PM20D1 activity between these two genotypes, suggesting that FAAH deficiency does not indirectly alter PM20D1 levels. We have added the following sentence to subsection “N-acyl amino acid metabolism in mice with selective FAAH blockade”: “We confirmed that PM20D1 activity is not altered in FAAH-KO plasma (Figure 4—figure supplement 1).”

We have now also determined that NATs are not secondary PM20D1 substrates or inhibitors by direct in vitro enzymatic assay. As shown in Figure 3—figure supplement 1, PM20D1 hydrolyzes only C20:4-Gly, but not C20:4-NAT or anandamide. Furthermore, the presence of C20:4-NAT does not inhibit PM20D1 activity, as measured by C20:4-Gly hydrolysis. To reflect these new data, we have added the following sentences in subsection “PM20D1 and FAAH exhibit overlapping but distinct substrate specificity in vitro”: “Conversely, PM20D1 exhibited robust hydrolysis activity only on C20:4-Gly (0.6 nmol/min/mg) and was unable to hydrolyze either anandamide or C20:4-NAT (Figure 3—figure supplement 1A). We also determined that C20:4-NAT was not a directly inhibitor for PM20D1 (Figure 3—figure supplement 1B).”

2) As a related point to #1, it would be useful to discuss (or measure) the rate of FAAH-mediated N-acyl taurine hydrolysis compared to that for N-acyl amino acid hydrolysis, to put this new FAAH activity in context.

We have now compared FAAH amidase activity on anandamide, C20:4-NAT, and C20:4-Gly side by side and included these data in Figure 3—figure supplement 1. Our data show that FAAH exhibits most robust activity on anandamide and lower, but still comparable activities on C20:4-NAT and C20:4-Gly. We have now discussed these new results in subsection “PM20D1 and FAAH exhibit overlapping but distinct substrate specificity in vitro”: “To better understand FAAH-mediated N-arachidonoyl glycine hydrolysis activity in the context of its previously described amidase activities, we directly compared the hydrolytic activity of transfected FAAH on C20:4-Gly, anandamide (C20:4-NAE), and N-arachidonoyl-taurine (C20:4-NAT). As expected, FAAH showed robust hydrolysis activity with anandamide as a substrate (0.5 nmol/min/mg) and lower, but similar activities with C20:4-NAT and C20:4-Gly (0.04 and 0.02 nmol/min/mg, respectively, Figure 3—figure supplement 1A).”

3) The activity of PM20D1 appears high in tissue (i.e. liver and kidney), though it is discussed as a secreted enzyme? Can the authors clarify this?

We agree with the reviewers that this point of confusion requires clarification. Many other enzymes secreted into the blood can also be directly detected in tissue homogenates, with one of the best examples being lipoprotein lipase (LPL). LPL activity in adipose and muscle tissues likely reflects the association of this enzyme to the endothelial cell surfaces and extracellular matrix within the tissues. To clarify our finding of PM20D1 activity in tissue lysates, we have added the following sentence to subsection “Detection of a second, PM20D1-independent N-acyl amino acid hydrolysis activity”: “The presence of PM20D1 activity in tissue homogenates reflects potential interactions of PM20D1 with the extracellular matrix or with cell surfaces, as has previously been observed with lipoprotein lipase and other secreted enzymes (Cryer, 1981).”